# Perineal Massage during Pregnancy for the Prevention of Postpartum Urinary Incontinence: Controlled Clinical Trial

**DOI:** 10.3390/medicina58101485

**Published:** 2022-10-19

**Authors:** María Álvarez-González, Raquel Leirós-Rodríguez, Lorena Álvarez-Barrio, Ana F. López-Rodríguez

**Affiliations:** 1Faculty of Health Sciences, University of León, Astorga Ave. 15, 24401 Ponferrada, Spain; 2SALBIS Research Group, Faculty of Health Sciences, University of León, Astorga Ave. 15, 24401 Ponferrada, Spain

**Keywords:** musculoskeletal manipulations, primary prevention, perineum, obstetric labor complications, physical therapy modalities

## Abstract

*Background and objectives*: Urinary incontinence is any involuntary loss of urine. It may result in anxiety, depression, low self-esteem and social isolation. Perineal massage has spread as a prophylactic technique for treating complications during labor. Acknowledged effects of perineal massage are reduction of incidence and severity of perineal tear and use of equipment directly related to the intrapartum perineal trauma. The aim of this study was to determine the effectiveness of massage in urinary incontinence prevention and identification of possible differences in its form of application (self-massage or by a physiotherapist), with the previous assumption that it is effective and that there are differences between the different forms of application. *Materials and Methods*: A controlled clinical trial with a sample of 81 pregnant women was conducted. The participants were divided into three groups: a group that received the massage applied by a specialized physiotherapist, another group that applied the massage to themselves, and a control group that only received ordinary obstetric care. *Results*: No differences were identified in the incidence or severity of urinary incontinence among the three groups. The severity of the incontinence was only affected by the body mass index and the weight of the baby at the time of delivery. *Conclusions*: A relationship between perineal massage interventions and development of urinary incontinence has not been observed.

## 1. Introduction

Urinary incontinence (UI) is defined by the International Continence Society as any involuntary loss of urine [1]. It may result in anxiety, depression, low self-esteem and social isolation [2]. There are three types of UI: stress UI—loss of urine upon effort or physical exertion (jumping) or sneezing or coughing due to the lacking capacity of the musculoskeletal system to compensate the increase in intra-abdominal pressure; urgency UI—loss of urine associated with a sudden and urgent need of urinating due to spasms of the bladder detrusor muscle; and mixed UI, when the patient shows a combination of both stress and urgency UI [3,4]. All cases are an important medical, social and economic problem that has an impact of the quality of life of women and that shows a growing tendency (though their prevalence is very different among different geographic regions and dependent on age) [5].

In recent years, perineal massage has spread as a prophylactic technique for com-plications during labor [6]. Acknowledged effects of perineal massage are reduction of incidence and severity of perineal tear and use of equipment directly related to the intrapartum perineal trauma [7,8]. In addition, the incidence of the two conditions (perineal tear and use of equipment) has been associated with the duration of delivery (especially the second phase). This has been related to the greater incidence of postpartum complications [9,10].

Therefore, we conclude that perineal massage has preventive effects on UI development in postpartum. The present investigation was consequently considered necessary in order to determine the effectiveness of massage in UI prevention and identification of possible differences in its form of application (self-massage or by a physiotherapist).

## 2. Materials and Methods

### 2.1. Design and Sample

A controlled non-randomized study was conducted with a sample of women selected from the maternity unit of their primary care center (first obstetric consultation with matron and/or gynecologist or through information leaflets, handed at the care attention center). Recruitment was carried out through three primary care centers served by the same hospital for six months. The women were selected through their interest in participating in the study by the information provided to them in their Primary Care Center (first obstetric consultation with a midwife and/or gynecologist or by notice through information brochures in their own clinic).

The sample was calculated using the G Power software package. The effect size was set as 0.5; α = 0.05; sample size = 81; actual power = 0.95 or 95%.

The inclusion criteria for participation were the following: (a) being aged between 18 and 40; (b) expecting a full-term delivery (week 37 or later); (c) expecting a singleton pregnancy; (d) expecting uncomplicated gestation and delivery; (e) not participating in any other psychoprophylaxis intervention; (f) delivering at Hospital Nuestra Señora de Sonsoles (Spain); (g) giving informed consent of participation in the study and attending to all intervention and or evaluation sessions. The exclusion criteria defined were: (a) any counter indication for perineal massage; (b) medical diagnosis of any pelvi-perineal pathology prior to becoming pregnant; (c) any records on cesarean delivery (in the present or previous deliveries); and (d) presence of UI prior to delivery identified through International Consultation on Incontinence Questionnaire-Short Form (ICIQ-SF) [11].

The sample consisted of 81 women (Figure 1). Throughout the course of the investigation, there was no sample loss.

### 2.2. Procedure

The research protocol was registered in ClinicalTrials.gov (ID: NCT05114811) on 10 November 2021. Participants were divided into three groups according to their personal preference: perineal massage (*n* = 27); self-massage group (*n* = 27), and a control group (*n* = 27) which had regular obstetric attention (regular medical control and information sessions with matron).

All participants signed an informed consent form according to Declaration of Helsinki (rev. 2013) and were also informed on the confidentiality of their personal data. Study had been previously approved by the Ethics Committee of the University of León, Spain (code: 021-2018).

Data collection took place in one evaluation session on the 5th or 6th postpartum week through a self-informed questionnaire where participants registered: characteristics of delivery (week of gestation, weight of the baby, duration of labor, posture, tear, episiotomy, use of equipment and/or analgesia), quality of life through the King’s Health Questionnaire (KHQ), and UI incidence through ICIQ-SF (punctuation higher than 0) and description (quantity of loss of urine and how this affects to their daily life), identified in the items included in the questionnaire.

The ICIQ-SF is a condition-specific questionnaire that assesses the subjective symptoms and quality of life of women with urinary incontinence [12,13]. The questionnaire consists of four items pertaining to the frequency of leakage, amount of leakage, interference with everyday life, and the perceived cause of leakage. For the first three questions, the patients were asked to rate their answers on a Likert scale where the maximum scores possible (corresponding to the greatest severity of the condition) were 5, 6, and 10, respectively. For the last question, the purpose of which was to diagnose the type of incontinence, the patients were asked to indicate all the circumstances under which urine leakage occurred. The scores for the frequency of leakage, amount of leakage, and interference with everyday life were added up to obtain the total score. The total score ranged from 0 to 21, and the higher the score, the more severe the condition [14]. The internal consistency of the Spanish version of this questionnaire is 0.89 for its Cronbach’s alpha score [15].

Finally, the KHQ is a condition-specific questionnaire that assesses the quality of life of women with urinary incontinence [16]. The KHQ consists of 21 items in the following nine domains: general health perceptions, incontinence impact, role limitations, physical limitations, social limitations, personal relationships, emotions, sleep/energy, and incontinence severity. Each KHQ domain provided a score, and the scores ranged from 0 to 100, the higher scores indicating poorer quality of life. The internal consistency of the different dimensions included in this questionnaire as well as its complete structure in its Spanish version is Good (0.65 < Cronbach’s alpha score > 0.92) [17].

### 2.3. Interventions Applied

(a)Perineal self-massage intervention. As described in a previous publication [16], self-massage group received standing instructions on perineal massage during pregnancy: it should be performed at least twice a week (on alternate days) for 10 min using a water-base lubricant from the 34th gestation week until delivery.(b)Perineal massage intervention. Perineal massage was applied by a physiotherapist expert in Urogynecol. and obstetrics over 6–10 sessions (from 34th gestation week until delivery) of 30 min each on a weekly basis. The intervention protocol has been used previously [16]. The procedure included direct manual techniques, the use of the EPI-NO^®^ device (Northampton, UK), and another external manual technique [18].

### 2.4. Statistical Analysis

The statistical analysis was carried out by an investigator blinded to experimental groups (unaware of the meaning of codification of the database of the three sample subgroups). The sample was described by descriptive statistical descriptions (frequency, percentages, media and typical deviation).

Kolmogorov–Smirnov tests and Levene’s test for equality of variances were applied to check the distribution of the data for the pretreatment measure of the outcome variables in the three experimental conditions. Since the results confirmed normal distribution and equality of variances, in categorical variables independent samples Chi-square test and Fisher exact test were used to verify the homogeneity of the groups, using Cramer’s V as measure of the effect sizes. The three groups’ repeated measure analyses of variance (ANOVA) were used to assess changes in clinical variables and psychosocial functioning, computing pairwise differences using Bonferroni correction, and partial eta-squared (η^2^_p_) was calculated to assess effect sizes. All effect sizes were interpreted using the benchmarks provided by Cohen [19] (η^2^_p_: small <0.06, medium >0.06 and <0.14, and large >0.14; Cramer’s V: small <0.3; medium >0.3 and <0.6, and large >0.6).

A correlation analysis was conducted between the impact of the quality of life due to UI and the quantity of loss of urine and other obstetric variables to find out the relationship between them. Moreover, we applied linear regression models using both the dependent variable and the independent, obstetric variables adjusted by age. R^2^ statistic was used to evaluate the fit in the linear regression models. Omega squared was calculated to evaluate the size of the effect of the models. All calculations were performed using the STATA software v.13 (Stata Corp., College Station, TX, USA). The significance level was set at *p* < 0.05.

## 3. Results

Descriptive analysis of the sample (Table 1) identified significative differences in the age variable between the control and massage subgroups (*p* < 0.01; η^2^_p_ = 0.11). Obstetric characteristics (Table 2) were statistically different only in relation to the incidence of episiotomy (X^2^ = 23; *p* < 0.001; V = 0.53).

A correlation analysis was conducted between the duration of labor and the weight of the baby and did not result statistically significative (*p* > 0.05). The association of quality of life and age, weight of the mother, body mass index (BMI), weight gained during pregnancy, duration of labor (*p* > 0.05) was also analyzed. Quality of life was only significantly correlated inversely with the week of delivery (r = −0.6; *p* = 0.006) and the weight of the baby (r = −0.6; *p* = 0.005) and directly with the quantity of urine loss (r = 0.7; *p* = 0.005). Quantity or loss of urine was only correlated to the BMI (r = 0.6; *p* = 0.03).

The linear regressions used as dependent variables were the following: number of deliveries, BMI, weight gained during pregnancy, week of delivery, severity of tear, duration of labor and weight of the baby (Table 3). The quality of life was only affected by the week of delivery and the weight of the baby at the time of delivery (−0.61 < B > −0.25; *p* < 0.01; 0.25 < ω^2^ > 0.26) and the quantity of urine loss was only affected by the BMI and the weight of the baby at the time of delivery (B = 0.04; *p* < 0.05; 0.04 < ω^2^ > 0.2).

## 4. Discussion

The objective of the present investigation was to determine the effectiveness of massage in UI prevention and identification of possible differences between its mode of application (self-massage or by a physiotherapist). After analysis of the results obtained, this intervention seems to have no influence on UI postpartum nor the severity of its consequences.

Prevalence of postpartum UI was not related with the distinction among the sample subgroups. This phenomenon was also identified by Eason et al. [20] with the peculiarity that it was identified three months postpartum; in the present study, such relation was discarded even in the early puerperium. Though perineal massage has been associated with a lower incidence in and use of equipment in labor, and these two phenomena have been repeatedly related to UI prevalence [21,22], we have not been able to establish a relationship between applied interventions and development of UI in the present study.

Severity of UI was also associated with the mother’s BMI (registered during the first trimester of pregnancy). This association is consistent with previous publications and has been thoroughly studied [23,24]. Previous studies have associated the weight of the baby with the incidence of UI postpartum [25,26], but this is the first investigation that associates it only with severity.

In the massage group, we identified that an incidence of episiotomy was statistically lower than in the control group and the self-massage group. However, no relevant differences were identified between the latter two groups. Such findings were consistent with previous investigations which had already identified the preventive effect of perineal massage on episiotomy [27,28,29], although this is the first time that a lower efficiency of auto-applied massage is stated, despite the fact that auto-applied massage has been a popular recommendation of professional obstetricians, especially during the recent months when health care assistance was generally not on-site due to the COVID-19 pandemic [30].

Duration of labor was not significantly associated with the weight of the newborns; this contradicts what has been published by a recent paper [31]. In any case, it is an association which has hardly been investigated in scientific literature. It has been established that active labor lasts for less than 12 h [32]; the only group that lasted longer was the self-massage group, and the group that lasted the least amount of time on delivery on average was the control group. Consequently, the present study discarded the influence of perineal massage on duration of delivery, regardless of its application mode. Therefore, the physiological mechanism by which perineal massage reduces the development of intrapartum perineal trauma appears to have no correlation with its influence on duration of delivery.

The authors must recognize that this research has methodological limitations. Fundamentally, the sample size (although representative and with proven statistical power) is small. Furthermore, the fact that the information analyzed was obtained through questionnaires means that the data could be biased or unreliable. It would have been an added value to the present investigation to include long-term follow-up of the characteristics of the participants’ urine leakage. Finally, the authors must acknowledge the implicit bias in the way the women were divided into the three subgroups: the fact that the women themselves decided whether or not to undergo perineal intervention could mask some extraneous variable that was not taken into account.

In any case, this investigation also presents strengths such as being the first with this research objective and containing a multitude of obstetric variables that act as extraneous variables not considered on other occasions. Furthermore, it should be the fruit of future research to assess whether the interventions evaluated here could have more significant or better effects on maternal perineal health if applied for a longer period of time (not only in the last weeks of gestation).

## 5. Conclusions

We have not been able to establish a relationship between applied interventions and development of UI in the present study. In the massage group, we identified an incidence of episiotomy statistically lower than in the other two. However, there were no differences in the incidence of episiotomy between the control and self-massage groups.

The results presented should be taken into account by health care professionals specializing in obstetrics; though prepartum perineal massage has physical and psychological benefits for women, there is no evidence that such procedure decreases incidence of postpartum UI. It is necessary to carry out more investigations studying the specific effects and benefits of perineal massage during pregnancy.

## Figures and Tables

**Figure 1 medicina-58-01485-f001:**
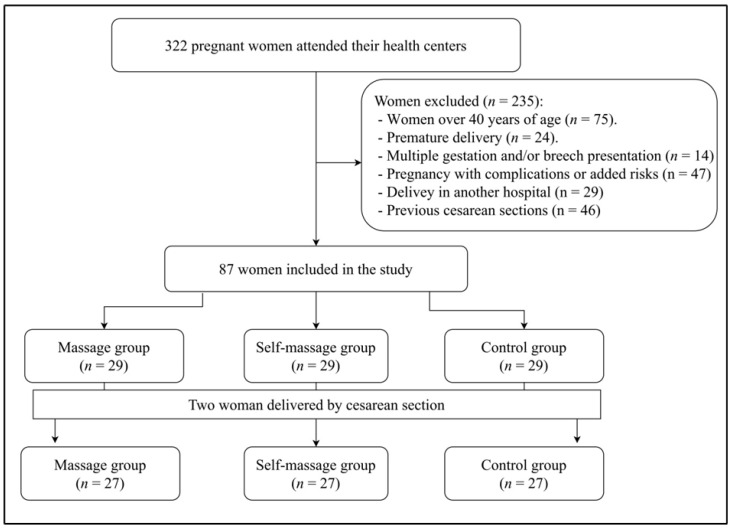
CONSORT flow diagram.

**Table 1 medicina-58-01485-t001:** Descriptive data of the sample (mean ± standard deviation).

	All (*n* = 81)	Control (*n* = 27)	Self-Massage (*n* = 27)	Massage (*n* = 27)
Age (years)	32.6 ± 4	30.7 ± 4.3 ^a^	33.2 ± 3.2	33.8 ± 3.8 ^a^
Height (cm)	164 ± 6.2	163.4 ± 6.3	163.8 ± 5.5	164.7 ± 6.9
Weight (kg)	58 ± 8.3	58.2 ± 9.3	59.1 ± 8.9	56.8 ± 6.5
Body Mass Index (kg/m^2^)	21.6 ± 2.8	21.8 ± 2.8	22 ± 2.9	21.2 ± 2.9
Weight gain (kg)	12 ± 4	12.6 ± 4.8	11.9 ± 3.4	11.5 ± 3.8
Deliveries (nº)	1.3 ± 0.5	1.4 ± 0.5	1.2 ± 0.4	1.4 ± 0.6
Labor week (nº)	39.3 ± 1.7	39 ± 2.3	39.4 ± 1.6	39.5 ± 1.2
Baby weight (kg)	3.3 ± 0.4	3.3 ± 0.5	3.2 ± 0.2	3.4 ± 0.3
Duration of labor (hours)	10.9 ± 8.1	9.3 ± 6.6	13 ± 9.3	10.4 ± 8.2

ANOVA significant results: ^a^ *p* < 0.05; control vs. massage.

**Table 2 medicina-58-01485-t002:** Delivery and urinary incontinence characteristics [data provided: *n* (percentage)].

	All (*n* = 81)	Control (*n* = 27)	Self-Massage (*n* = 27)	Massage (*n* = 27)
Episiotomy *	35 (43.2%)	19 (70.4%)	14 (51.9%)	2 (7.4%)
Perineal tear:				
No	57 (70.4%)	15 (55.6%)	20 (74.1%)	22 (81.5%)
Mild	17 (21%)	8 (29.6%)	5 (18.5%)	4 (14.8%)
Moderate/severe	7 (8.6%)	4 (14.8%)	2 (7.4%)	1 (3.7%)
Position:				
Lithotomy	63 (77.8%)	25 (92.6%)	21 (77.8%)	17 (63%)
Sideways	5 (6.2%)	1 (3.7%)	3 (11.1%)	1 (3.7%)
Sit/squat	11 (13.6%)	1 (3.7%)	2 (7.4%)	8 (29.6%)
Standing	2 (2.5%)	0 (0%)	1 (3.7%)	1 (3.7%)
Instrumental:				
No	64 (71.9%)	17 (63%)	23 (85.2%)	24 (88.9%)
Vacuum	10 (12.4%)	6 (22.2%)	1 (3.7%)	3 (11.1%)
Forceps	7 (8.6%)	4 (14.8%)	3 (11.1%)	0 (0%)
Analgesia:				
No	16 (19.8%)	4 (14.8%)	5 (18.5%)	7 (25.9%)
Local	2 (2.5%)	1 (3.7%)	1 (3.7%)	0 (0%)
Epidural	63 (77.8%)	22 (81.5%)	21 (77.8%)	20 (74.1%)
Urinary incontinence:				
No	56 (69.1%)	18 (66.7%)	15 (55.6%)	23 (85.2%)
Yes	25 (30.9%)	9 (33.3%)	12 (44.4%)	4 (14.8%)
Severity of urinary incontinence (perception of amount of urine from leaks):				
Nothing	56 (69.1%)	18 (66.7%)	15 (55.6%)	23 (85.2%)
Little	24 (29.6%)	8 (29.6%)	12 (44.4%)	4 (14.8%)
Moderate	1 (1.2%)	1 (3.7%)	0 (0%)	0 (0%)
A lot	0 (0%)	0 (0%)	0 (0%)	0 (0%)
Quality of life (mean ± standard deviation):				
0–100 points	39.6 ± 20.1	31.4 ± 34.1	57.3 ± 13.4	50 ± 18.9

Chi-squared significant results: * *p* < 0.001.

**Table 3 medicina-58-01485-t003:** Linear regression models of impact of urinary incontinence on quality of life and severity of urinary incontinence in relation to obstetrics variables (continuous variables) adjusted by age.

Variable	Quality of Life			UI Severity		
	B	SE	R^2^	B	SE	R^2^
Number of deliveries	−0.27	1	0.003	0.02	0.11	0.001
Body Mass Index	0.02	0.18	0.001	0.04 *	0.02	0.05
Weight gain	0.04	0.16	0.003	0.01	0.01	0.002
Labor week	−0.61 **	0.2	0.286	−0.01	0.03	0.99
Perineal tear	−0.97	0.99	0.04	0.08	0.121	0.005
Duration of labor	0.01	0.06	0.002	0.01	0.007	0.002
Baby weight	−0.25 **	0.01	0.3	0.04 **	0.001	0.005

UI: urinary incontinence; B: regression coefficient; SE: standard error; R^2^: coefficient of determination. * *p* < 0.05; ** *p* < 0.001.

## Data Availability

The dataset used and analyzed during the current study are available from the corresponding author.

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
