# Peer review of "Perineal Massage during Pregnancy for the Prevention of Postpartum Urinary Incontinence: Controlled Clinical Trial"

_medicina, 2022, doi:10.3390/medicina58101485_

Round 1

Reviewer 1 Report

This article aims to determine the effectiveness of massage in urinary incontinence prevention and identification of possible differences in its form of application (self-massage or by a physiotherapist). The subject of the manuscript is interesting, current and needed nowadays. Firstly, I thank the authors for including the calculation of the power of the sample size and the effect sizes. However, there are some minimal considerations for improving readability and scientific strength of this manuscript:

11) Authors should update the references (half of the references are over 10 years old).

22) Internal consistency of the questionnaires should be reported by calculating traditional coefficients such as Cronbach's alpha and/or McDonald's omega.

33) The limitations are very vaguely described. Please clarify the implications of not having randomized the trial.

44) There are some spelling errors, please check.

Nonetheless, it is a potentially important contribution to the literature. Further, I believe that this article will be useful for the readers of Medicina.

Author Response

Dear Editor and Reviewer of Medicina:

Thank you very much for your suggestions and contributions to improve the quality of the manuscript. Following your indications, we respond, point by point, to the reviewers' comments.

In the text, all the modified or added sentences have been written in red to facilitate the correction by the reviewers.

1) Authors should update the references (half of the references are over 10 years old).

The authors have substituted as many references older than ten years as possible (ensuring the quality and accuracy of the content of the works cited). In total, we have included nine new references

2) Internal consistency of the questionnaires should be reported by calculating traditional coefficients such as Cronbach's alpha and/or McDonald's omega.

The authors have added this information for the two questionnaires used in this research.

3) The limitations are very vaguely described. Please clarify the implications of not having randomized the trial.

The authors have expanded the limitations section following your advice.

4) There are some spelling errors, please check.

The text has been thoroughly proofread by an expert translator and native English speaker.

Once again, thank you very much for the time spent and the interest shown in this work; as well as in the positive evaluations you have given of it.

Receive a warm greeting,

The authors.

Reviewer 2 Report

I consider only one point that could have been discussed by the authors. Would consistent perineal massage during the 3rd trimester have different results from those observed in this study?

Author Response

Dear Editor and Reviewer of Medicina:

Thank you very much for your suggestions and contributions to improve the quality of the manuscript. Following your indications, we respond, point by point, to the reviewers' comments.

In the text, all the modified or added sentences have been written in red to facilitate the correction by the reviewers.

1. I consider only one point that could have been discussed by the authors. Would consistent perineal massage during the 3rd trimester have different results from those observed in this study?

The authors have added a sentence at the end of the Discussion that refers to what you indicate.

Once again, thank you very much for the time spent and the interest shown in this work; as well as in the positive evaluations you have given of it.

Receive a warm greeting,

The authors.